# A Systematic Review of PET Contrasted with MRI for Detecting Crossed Cerebellar Diaschisis in Patients with Neurodegenerative Diseases

**DOI:** 10.3390/diagnostics13101674

**Published:** 2023-05-09

**Authors:** Shailendra Mohan Tripathi, Naif Ali Majrashi, Ali S. Alyami, Wael A. Ageeli, Turkey A. Refaee

**Affiliations:** 1Department of Geriatric Mental Health, King George’s Medical University, Lucknow 226003, India; 2Diagnostic Radiography Technology (DRT) Department, Faculty of Applied Medical Sciences, Jazan University, Jazan 85145, Saudi Arabia

**Keywords:** dementia and related disorders, neurodegenerative disorders, neuroimaging, nuclear medicine, Parkinson disease and related disorders

## Abstract

There has not been extensive research into crossed cerebellar diaschisis (CCD) in neurodegenerative disorders. CCD is frequently detected using positron emission tomography (PET). However, advanced MRI techniques have come forth for the detection of CCD. The correct diagnosis of CCD is crucial for the care of neurological patients and those with neurodegenerative conditions. The purpose of this study is to determine whether PET can offer extra value over MRI or an advanced technique in MRI for detecting CCD in neurological conditions. We searched three main electronic databases from 1980 until the present and included only English and peer-reviewed journal articles. Eight articles involving 1246 participants met the inclusion criteria, six of which used PET imaging while the other two used MRI and hybrid imaging. The findings in PET studies showed decreased cerebral metabolism in the frontal, parietal, temporal, and occipital cortices, as on the opposite side of the cerebellar cortex. However, the findings in MRI studies showed decreased cerebellar volumes. This study concludes that PET is a common, accurate, and sensitive technique for detecting both crossed cerebellar and uncrossed basal ganglia as well as thalamic diaschisis in neurodegenerative diseases, while MRI is better for measuring brain volume. This study suggests that PET has a higher diagnostic value for diagnosing CCD compared to MRI, and that PET is a more valuable technique for predicting CCD.

## 1. Introduction

Neurodegenerative diseases, which overwhelm the health care system, are described as the progressive loss and dysfunction of vulnerable neurons, synapses, glial cells, and their networks [1,2] and contrast with static neuronal loss due to toxic or metabolic disorders. The accumulation or deposition of proteins with toxic effects in the nervous system, particularly in the neurons and glial cells, leads to neurodegenerative diseases [2]. Neurodegenerative diseases are classified based on their clinical presentations, affected anatomical regions, altered proteins involved in the pathogenetic processes, and aetiology. The most common forms of neurodegeneration include amyloidoses, tauopathies, alpha-synucleinopathies, and transactivation response DNA-binding protein 43 (TDP-43) proteinopathies [2]. The most frequent form of amyloidosis is Alzheimer’s disease (AD). It is the most common form of dementia and is characterized by a temporally and spatially predictable pattern of synaptic dysfunction and neuronal death due to the progressive accumulation of intracellular neurofibrillary tangles and extracellular amyloid (A) plaques in the cerebral cortex [3]. Lewy body dementia (DLB) accounts for 10–15% of all cases of late-onset dementia and is the second most common and pervasive neurodegenerative form of dementia in the elderly [4]. DLB’s clinical signature is the aggregation of abnormal α-synuclein in Lewy bodies and neurites. Neurofibrillary tangles and neuritic plaques on autopsy reveal that DLB often coexists with AD [5]. Nigrostriatal dopaminergic neurodegeneration is a hallmark of DLB, making dopaminergic imaging a promising diagnostic technique for distinguishing this disorder from AD [3]. The list of coexisting neurodegenerative conditions may be much longer than expected, and it is really difficult to get pure neurodegenerative pathologies [5]; therefore, exploration of other phenomena that have the potential to change future diagnostic and therapeutic strategies is urgently needed. Depressed regional metabolism in neurodegenerative conditions on functional neuroimaging techniques is an indicator of functional impairment related to specific locations in the brain, and the presence of a similar depressed regional metabolism in anatomically separate but functionally related regions such as the cerebral and cerebellar hemispheres open a new avenue of research [6,7]. The presence of functional connections in anatomically separate brain locations can be seen through imaging techniques, and one such phenomenon appreciated in various neurological conditions, including neurodegenerative disorders, is crossed cerebellar diaschisis (CCD).

Diaschisis is a neurological condition that was initially introduced in 1914. It pertains to the state of secondary neural depression in the brain, which arises from the loss of connections to injured neural regions that are distant from the actual brain lesions. These regions may include the ipsilateral cerebral cortex [8] or contralateral cerebellum [9]. This phenomenon is commonly referred to as CCD [5]. CCD is defined as a well-recognized phenomenon characterized by a decrease in both metabolism and blood flow within the cerebellar or unilateral cerebellar hemisphere that is opposite to a focal supratentorial lesion or a damaged supratentorial area [10,11]. Franceschi and colleagues have also defined CCD as an imaging artifact that is observed in patients who have suffered from various supratentorial insults, including traumatic brain injury and cerebral infarcts [6]. It has been suggested that these insults result in the destruction of corticobulbar, corticopontine, and corticocerebellar fibre tracts [6]. The precise mechanism of CCD is widely unknown, but it is hypothesized that CCD involves disruption of the cortico-ponto-cerebellar fibres [11]. The pathophysiology of CCD has been thought to possess a functional or reversible underpinning and may encompass a secondary morphological alteration, suggesting a prognostic indicator of neurological improvement and clinical outcomes [12]. However, trans-neuronal degeneration of the cortico-ponto-cerebellar pathway has also been suggested, which results in cerebellar degeneration in response to contralateral cerebral pathology and has implications for the functional basis and reversibility of the phenomenon [13]. There are pieces of evidence to demonstrate the association of the CCD phenomenon with several acute neurological conditions, including stroke and other forms of brain injury such as a supratentorial tumor, epilepsy, radiation necrosis, and encephalitis [10]. Further, CCD has also been associated with a degree of hemiparesis at clinical examination, as well as functional impairment and disconnection [14,15]. Traditionally, CCD is thought to be a phenomenon observed in acute conditions in response to a focal supra-tentorial lesion, but lately, it has been observed in neurodegenerative disorders which primarily involve the supra-tentorial cerebral cortex. Notably, over the last few years, researchers have observed CCD phenomenon in chronic neurological conditions including Alzheimer’s disease [7,11,16]. The occurrence of CCD in AD can be due to the early involvement of cortico-ponto-cerebellar (CPC) tracts in the course of the disease, which was not very well explained earlier by researchers, as Alzheimer’s disease was thought to be predominantly a disease of the cerebral cortex with cerebellar involvement very late in the disease process. CPC tracts originate from the cerebral cortex and reach the contralateral cerebellar cortex by descending through synapses in anterior pontine nuclei, and demonstration of CCD phenomenon in the initial stages of AD points towards early involvement of the cerebellum in AD [7,17]. AD is a neurodegenerative condition which predominantly affects the cerebral cortex, and demonstration of cerebellar cortex involvement early in the course of the disease through CCD phenomenon in such a gradually progressive condition warrants further exploration in order get insight into the aetio-pathogenesis of the disease. 

Due to the development of new imaging techniques, CCD has become a well-recognized phenomenon. There are three types of neuroimaging techniques used to detect CCD, including positron-emission tomography (PET), single photon-emission computed tomography (SPECT), and magnetic resonance imaging (MRI) perfusion techniques, especially the arterial spin labelling perfusion MRI (ASLPWI). On visual inspection of 18F-Fluorodeoxyglucose (FDG) brain investigations, CCD is thought to be a prevalent side effect [18]. The ASLPWI is used to detect CCD in supra-tentorial intracerebral hemorrhage and ischemic infarction [19], ischemic stroke [20], after brain tumor surgery [21], and epilepsy [22]. The ASLPWI is a perfusion technique that uses magnetically tagged blood water as an endogenous tracing method to quantitatively assess cerebral blood flow (CBF) without the injection of contrast agents [23].

Given that the phenomenon of CCD has been widely studied in several neurological diseases but sparsely in neurodegenerative conditions using imaging techniques, we have limited knowledge of the prevalence of CCD in neurodegenerative conditions and the potential impact of CCD on the progression of neurodegeneration. CCD is traditionally diagnosed by molecular imaging techniques and is often incidentally detected on PET images. However, recent studies using advanced techniques in MRI such as ASL-DWI, DWI, and DTI are frequently published to detect CCD instead of functional imaging such as FDG-PET studies [10,13]. It has been argued that for neurological patients, a precise diagnosis of CCD can have significant ramifications, especially given that the current gold standard for diagnosis, PET, is often not conducted until after acute cortical pathology has developed [13]. Therefore, the value of imaging techniques (MRI or SPECT/PET) in diagnosing CCD in neurological conditions is not yet well established. Two broad questions were formulated to guide our systematic review:Are there any imaging modalities other than FDG-PET to detect CCD?What is the most accurate imaging technique used to detect CCD in neurological diseases?

The purpose of this article is to give a thorough systematic analysis of existing imaging research, particularly MRI, PET, and SPECT studies, detecting CCD in neurological conditions and then determine which technique could be more valuable, specific, and better used in the diagnosis of CCD.

## 2. Methods

The Preferred Reporting Items for Systematic Reviews and Meta-analyses (PRISMA) guidelines served as the basis for our search strategy [24]. Titles were screened from databases using the PRISMA chart (figure) and the PICO (P = populations/people/patient/problem, I = intervention(s), C = comparison, and O = outcome) worksheet and search technique were used to determine eligibility for further analysis (See the study framework section). The search was expanded to incorporate topic-specific phrases. The selected papers’ full texts were obtained for in-depth reading and data extraction in accordance with the study’s aims and goals. The current systematic review was previously registered in the PROSPERO database (https://www.crd.york.ac.uk/prospero/) accessed on 22 May 2022, prior to the analysis being undertaken (registration number CRD42022329424). 

### 2.1. Data Sources and Search Strategy

Online searches of the databases MEDLINE: Scopus, PubMed, Embase, and Primo (used by the University of Aberdeen) were performed in April–September 2022 (the last search was carried out on 15 September 2022). These online databases were used to find literature relevant to the detection of CCD in neurodegenerative diseases by imaging, including SPECT, PET or FDG-PET, and MRI. We searched for specific words in the title, abstract, and MESH (Medical Subject Headings) terms using the eligibility and exclusion criteria. Keywords used to search for relevant literature were combined (“crossed cerebellar diaschisis”, OR Diaschisi (MeSH)” OR CCD OR “Cerebellar diaschisis”), AND (“neurodegenerative diseases (MeSH)” OR “neurodegenerative disorders (MeSH)” OR “neurodegenerative conditions”), AND (Imaging OR SPECT OR PET OR “FDG-PET (MeSH)” OR MRI (MeSH)). The literature found to be relevant was saved for further review to extract important information for this study. To ensure that no relevant publications were overlooked, we also manually checked through all of the articles’ cited references.

### 2.2. Study Selection Criteria

Two authors (ST and NM) independently searched for relevant literature from the above highlighted online databases using the defined and highlighted search terms, as well as screened the title, abstract, and full text of identified articles in this systematic review. Based on the abstracts, the authors decided to read the full text, especially when there was any doubt. All relevant previous imaging studies detecting CCD in neurodegenerative diseases were downloaded, recorded, and analysed as per the aim of this systematic review. Only studies that satisfied the following conditions were considered:Published as an English journal article. Articles other than the English language were excluded. Because we restricted the search to English-published results, this may increase our susceptibility to publication bias.Used imaging techniques, including SPECT, PET, or FDG-PET, MRI, or any other imaging modality.Detected CCD in patients diagnosed with neurological diseases other than neurodegenerative diseases. Imaging studies that detected CCD in patients diagnosed with tumors, infarctions, etc., were excluded.

### 2.3. Data Extraction

The authors independently reviewed the titles, abstracts, and, where appropriate, full texts, of all retrieved references. When there was any ambiguity in the search process, the decision was made by consensus. We used a standardized data abstraction form to extract the data and tabulate the results of the included research; this contained the fundamental characteristics of the included studies, such as author, publication date, research type, number of patients, study design (retrospective or prospective), and main results and outcomes. We also extracted the basic characteristics of the research subjects or patients, such as age (mean age) and sex. Any discrepancies were solved by consensus.

### 2.4. Assessment of Study Quality

The Newcastle–Ottawa Scale for case-control studies, a modified version of the Newcastle–Ottawa Cohort Scale for cross-sectional studies [25], and a quality assessment tool for case series studies were used to independently evaluate the quality of the included studies or the risk of bias. The quality of the included cross-sectional studies was evaluated using a three-point scale. These included (a) selection, which includes representativeness of the sample, sample size, non-respondents, and ascertainment of exposure; (b) comparability, which includes an assessment of study design and analysis, as well as whether any confounding variables were adjusted for; and (c) outcome, which includes an assessment of the outcome ascertained by independent blind assessment, record linkage, or self-report, and a description of appropriate statistical tests. Using the Newcastle–Ottawa Scale, we ranked the quality of cross-sectional and case-control studies as “good”, “very good”, “satisfied”, or “unsatisfied” in each domain. To be regarded as “good”, quality scores had to fall within the range of 3–4 on the selection scale, 1–2 on the comparability scale, and 2–3 on the outcomes scale. Two stars for the selection, one or two for the comparability, and two or three for the outcomes constituted a “fair” quality rating. Poor quality was indicated by a score of 0 or 1 stars across the board (no stars for selection, 0 stars for comparability, and 0 or 1 stars for outcomes) (Table 1). Very high-quality studies were designated by a rating of 9–10 stars, good quality studies by a rating of 7–8 stars, satisfied studies by a rating of 5–6 stars, and poor-quality research by a rating of 0–4 stars. Case-series studies were also evaluated, and we gave them a rating of (good, fair, and poor). Please refer to Table 1 for further details. Disagreements were settled through mutual agreement. 

## 3. Results

### 3.1. Study Selection and Characteristics of Included Studies

The selection process for included studies is depicted in Figure 1. In short, a search of databases including PubMed, Scopus, Google Scholar, and Primo revealed 273 relevant studies. Then, seven papers were excluded because they were duplicates. A total of 2502 papers did not make the cut after automated techniques were used to analyse the titles, abstracts, and findings and it was determined that they were not relevant. Fourteen full-text articles were assessed for eligibility. Another six articles were removed for additional reasons. Finally, eight studies were included in our systematic review, involving 1246 participants (Table 2). Of these participants, 1170 were patients with different neurological disorders, while 76 were healthy controls. All studies were conducted in the USA except one study [7], which was conducted at multiple centres, and all were published between 1989 and 2022 (Table 2). All studies were cross-sectional and retrospective. Sample sizes ranged from 4 to 830 patients. All studies only enrolled patients diagnosed with neurodegenerative diseases using functional (SPECT or PET) or structural (MRI) imaging. The mean or median age of patients ranged from 22 to 90 years. More characteristics of eligible studies are summarized in Table 2 below.

### 3.2. Summary of Study Findings

The study results are summarized in Table 2. Out of eight studies included in this review, six studies were PET studies and used F^18^-FDG-PET, F^18^-FDG-PET/CT, 11C-PiB-PET, or ^18^F-Flortaucipir to visually detect CCD in patients with neurodegenerative diseases, while the remaining studies were MRI studies that used the ASL technique and hybrid imaging (PET/MRI). PET studies showed (a) a pattern of decreased cerebral metabolism in the frontal, parietal, temporal, and occipital cortices (lateralized findings), (b) crossed hypometabolism of the cerebellum, (c) a pattern of lower relative tracer delivery in the cerebellum, (d) 18F-FDG asymmetry reversal in the brain’s frontal and parietal lobes, as well as the basal ganglia, (e) negative cerebellar asymmetry indices (AI) but positive AI for thalamus, and (f) a decrease in the cerebellar regional cerebral blood flow (rCBF) and regional cerebral metabolic rate of oxygen metabolism (rCMRO2). In most studies, the PET findings were reported to be lateralized, with the left cerebral hemisphere being more affected than the right, whereas the right hemisphere of the cerebellum was more affected than the left. MRI and hybrid imaging studies showed (a) smaller whole brain volume and white matter volume in patients with CCD compared to patients without CCD, (b) reduced FDG uptake in the cerebellar hemisphere contralateral to the supratentorial cortical hypometabolism, and (c) characteristic imaging findings of frontotemporal dementia, semantic primary progressive aphasia, logopenic primary progressive aphasia, corticobasal degeneration, and AD dementia. Studies utilizing 18F-fluorodeoxyglucose in patients with neurodegenerative disorders like Alzheimer’s disease have shown promise, suggesting that PET is a sensitive approach for detecting crossed cerebellar as well as uncrossed basal ganglia and thalamic diaschisis. However, the results of MRI studies, which focused on brain morphology, may suggest that CCD occurs most commonly in frontal, temporal, and occipital dementia, as well as in patients with cortico-basal degeneration. Finally, CCD was detected in patients with Alzheimer’s (AD), AD dementia, progressive dementia, Lewy body dementia, mild cognitive impairment (MCI), aphasia, single nucleotide polymorphisms (SNP), and Amyotrophic lateral sclerosis (ALS) in both PET and MRI studies. 

### 3.3. Quality Assessment

Table 1 demonstrates the quality scores for the included studies, assessing the risk of bias. In brief, one study was of very good quality [18], four studies were of good quality [7,10,16,27], one study was of satisfied quality [6], one was of fair quality [28], and one study was of unsatisfied quality [11] due to a lack of enough information regarding sample size, representatives of the sample, non-participation rate, and statistical tests. Overall, almost two-thirds of the studies were considered to have good quality scores in all domains.

## 4. Discussion

There is a dearth of literature on CCD, a well-recognized phenomenon in neurological disorders, other than in neurodegenerative disorders, and we found eight studies exploring the phenomenon in neurodegenerative disorders. Most of the studies used the FDG-PET imaging technique to explore the phenomenon, and all of them were retrospective studies. Notably, six studies used FDG-PET, and two studies used MRI. Traditionally, the CCD phenomenon has been explored through PET imaging since its description in supra-tentorial brain infarction four decades ago [9]. Hypometabolism in the contralateral cerebellar cortex caused by supratentorial brain lesions is a reflection of cortico-ponto-cerebellar tract involvement and is thought to be more prevalent in asymmetric neurodegenerative conditions such as cortico-basal syndrome and the semantic variant of primary progressive aphasia [18]. However, the exploration of the CCD phenomenon in AD, traditionally involving a symmetrical brain [29], needs further explanation. Most of the studies included in this current systematic review have explored the CCD phenomenon in AD, and only a few studies have explored the phenomenon in other neurodegenerative disorders. Exploration of CCD, more so in AD, could be due to the higher prevalence of AD in comparison to other neurodegenerative disorders [30]; hence, the chances of more studies on the AD population. Further, the exploration of CCD through molecular imaging, especially 18F-FDG-PET, was noted in the selected studies, such as 18F-FDG-PET is better at detecting the CCD phenomenon in comparison to other imaging modalities, including MRI, irrespective of the nature of the supra-tentorial brain lesions [31].

The CCD phenomenon manifests as hypoxia and hypometabolism on the side opposite the injured supratentorial region, and the imaging modalities measuring brain metabolism and perfusion are ideal for its detection [32]. None of the studies in this systematic review used SPECT as a method of CCD detection. However, other imaging techniques measuring brain perfusion, including ASL-PWI were used in the studies for the detection of CCD and are included in this review. ASL-PWI was not only able to detect the CCD phenomenon in 33.8% of AD participants but also had the ability to identify structural differences in brain volumes and white matter volumes between AD participants with and without the CCD phenomenon [10].

None of the studies used only structural imaging, such as MRI, without measuring perfusion to detect CCD phenomena. The MRI measuring perfusion could detect the phenomenon in one-third of AD participants; however, one of the studies using FDG-PET could identify the CCD phenomenon in 830 AD participants [7,10]. Thus, functional imaging, which can detect metabolism or perfusion in the brain, can be considered superior in detecting the CCD phenomenon in comparison to structural imaging, and among functional imaging, PET, especially FDG-PET, was the most commonly used imaging technique to detect the CCD phenomenon in neurodegenerative disorders.

## 5. Conclusions

The CCD phenomenon is not a very new concept for acute neurological conditions such as infarcts and epilepsy, but it has recently drawn the attention of researchers in neurodegenerative disorders, especially AD. The phenomenon in neurodegenerative disorders was detected mostly through imaging techniques that can either detect metabolism or perfusion, such as PET, SPECT, or ASL-PWI, and amongst them FDG-PET is the most commonly used imaging technique for the detection of the CCD phenomenon.

## Figures and Tables

**Figure 1 diagnostics-13-01674-f001:**
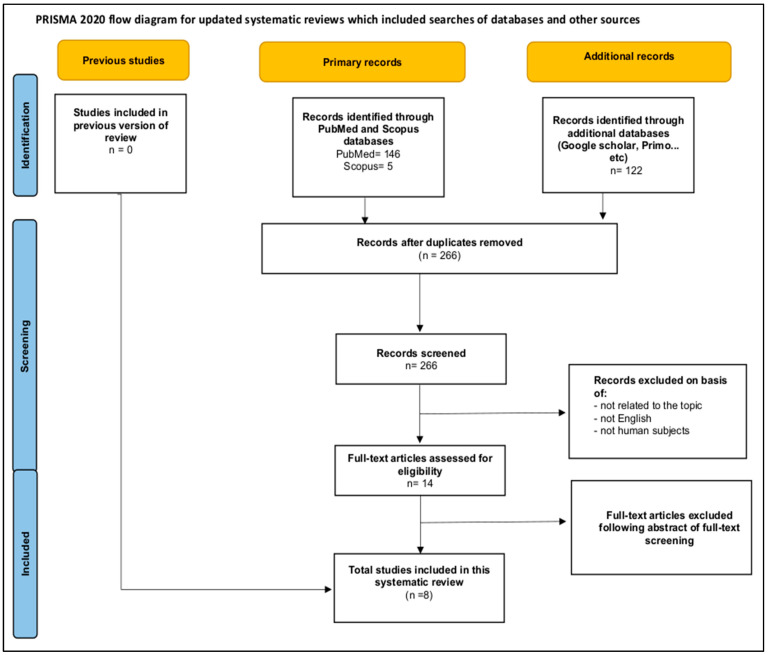
PRISMA flow chart showing the selection of appropriate studies.

**Table 1 diagnostics-13-01674-t001:** Risk of bias assessment (Newcastle–Ottawa Quality Assessment Scale criteria for case-control and cross-sectional studies) + case-study assessment.

**Study (First Author)**	**Study Design**	**Selection**					**Comparability**	**Outcome**		
		**Representativeness of the sample**		**Sample size**	**Non-respondents**	**Ascertainment of the exposure**	**Based on design and analysis**	**Assessment of outcome**	**Statistical test**	
[7]	Cross-sectional	+		+	+	++	-	++		
[10]	Cross-sectional	+			+	++	+	++	+	
[18]	Cross-sectional	+			+	++	++	++	+	
[6]	Cross-sectional	+				++	+	++		
[11]	Cross-sectional						-	++		
**Study (First author)**	**Study design**	**Selection**					**Comparability**	**Exposure**		
		**Case definition adequate?**		**Representativeness of the cases**	**Selection of controls**	**Definition of controls**	**Based on design and analysis**	**Ascertainment of exposure**	**Same method for cases and controls**	**Non-response rate**
[26]	Case-control	+		+	+	+	+	+	+	
[16]	Case-control	+		+	+	+	+	+	+	
**Study (First author)**	**Study design**	**Q1**	**Q2**	**Q3**	**Q4**	**Q5**	**Q6**	**Q7**	**Q8**	**Q9**
[27]	Case study	√	√	√	x	x	√	x	x	√

- Quality scores for Newcastle–Ottawa scale. Very good studies = 9–10 stars, good studies = 7–8 stars, satisfactory studies = 5–6 stars, and unsatisfactory studies = 0–4 stars [28]. For case-series quality scores: Q1: Was study question or objective clearly stated?, Q2: Was study population clearly and fully described, including case definition?, Q3: Were cases consecutive?, Q4: Were subjects comparable?, Q5: Was intervention clearly described?, Q6: Were outcome measures clearly defined, valid, reliable, and implemented consistently across all study participants?, Q7: Was length of follow-up adequate?, Q8: Were statistical methods well-described?, and Q9: Were results well-described? Good: met 7–9 criteria, Fair: met 4–6 criteria, Poor: met 0–3 criteria.

**Table 2 diagnostics-13-01674-t002:** Characteristics of enrolled studies detecting CCD in neurodegenerative diseases.

Authors	Sample Size	Age Range	Study Center (Site)	Imaging Modality	Parameters	Main Results	Study Design
(Tripathi et al., 2022 [7])	830 AD patients	Mean age = 70.72	Multicentric	[^18^F]-FDG-PET	18F-FDG-PET Quantification of FDG-PET hypometabolism was carried out using a standardized uptake value ratio (SUVR), with pons as the comparison region.	There were significant differences in the SUVR in different brain regions including temporal, occipital, and cerebellar cortices with right and left asymmetry. The SUVR was lower in the left temporal and occipital regions, whereas the SUVR was lower in the right side of the cerebellum.	Retrospective
(HERTEL et al., 2021 [10])	65 (43.1% females, 56.9% males)	51–88 (mean = 74)	NA	MRI (ASL PWI)	Pulsed ASL PWI with field of view 256 × 256, acquisition matrix 64 × 64, number of slices 9, slice thickness 8 mm, echo time 11 ms, repetition time 2500 ms, number of averages: 1, and duration 5:57 min.	A total of 65 patients were included in the study and 22 (33.8%) patients were found to be CCD-positive. CCD-positive patients had a significantly smaller whole brain volume (862.8 ± 49.9 vs. 893.7 ± 62.7 mL, and *p* = 0.049) and white matter volume (352.9 ± 28.0 vs. 374.3 ± 30.7, and *p* = 0.008) in comparison to CCD negative patients. Conclusion: ASL PWI was able to detect CCD in approximately one-third of patients with AD and was associated with smaller whole brain and white matter volume.	Retrospective
(Provost et al., 2021 [18])	197 (39% male)	22–90 (mean = 67)	At the Memory and Aging Center, University of California San Francisco (UCSF),	[^18^F]FDG-PET PET PET/CT	− All patients and controls underwent ^18^F-FDG brain PET on a PET/CT (Siemens, Erlangen, Germany) in 3D acquisition mode. − 117 patients also underwent scanning with ^11^C-PIB and ^18^F-Flortaucipir on the same scanner.	Cerebellar ^18^F-FDG asymmetry was associated with reverse asymmetry of ^18^F-FDG in the cerebral cortex (especially frontal and parietal areas) and basal ganglia. Significant CCD was present in 47/197 (24%) patients and was most frequent in corticobasal syndrome and semantic and logopenic variants of primary progressive aphasia. Mediation analysis in β-amyloid-positive patients had ^18^F-Flortaucipir cortical asymmetry and it was associated with cerebellar ^18^F-FDG asymmetry.	Retrospective
(Franceschi et al., 2021 [6])	75 (31 M, 44 F),	mean age = 74	Manhasset Images were obtained with an integrated 3-T PET/MRI system. PET surface maps, fused T1-weighted magnetization-prepared rapid acquisition gradient echo, and axial FLAIR/PET images were generated with a postprocessing software	Hybrid Imaging (FDG-PET/MRI), 2 blinded radiologists, and a nuclear medicine physician evaluated the PET/MRI images for pattern of neurodegenerative diseases.	IV injection FDG was administered for brain imaging in a 3-T PET/MRI system). A dual-echo T1-weighted gradient-recalled echo sequence was performed to acquire the MRI attenuation-correction map based on Dixon segmentation. The PET image matrix size was 344 × 344 × 127 mm, and transaxial voxel dimensions were 1.04 × 1.04 mm with a thickness of 2.03 mm.	Qualitative assessment showed that 10 of 75 (7.5%) patients had decreased metabolism in the cerebellar hemisphere contralateral to the supratentorial cortical hypometabolism consistent with CCD. Six of the ten patients had characteristic imaging findings of frontotemporal dementia (three behavioral variant frontotemporal dementia, two semantic primary progressive aphasia, and one logopenic primary progressive aphasia), three had suspected corticobasal degeneration, and one had Alzheimer dementia. Results of the study revealed that frontotemporal dementia, particularly the behavioral variant, and in patients with cortico-basal degeneration.	Retrospective
(Reesink et al., 2018 [11])	4 subjects	NA	NA	[^18^F]-FDG-PET&11C-PiB-PET	combination of 18F-FDG PET and 11C-PiB PET imaging	18F-FDG PET showed a pattern of cerebral metabolism with relative decrease in the frontal-parietal cortex of the left hemisphere and crossed hypometabolism of the right cerebellum. 11C-PiB PET showed symmetrical amyloid accumulation, but a lower relative tracer in the left hemisphere. Thus, the CCD could be explained on the functional basis due to neurodegenerative pathology in the left hemisphere. There was no structural lesion and the symmetric amyloid accumulation did not correspond with the unilateral metabolic impairment.	Retrospective study
(Akiyama et al., 1989 [16])	26 patients	NA	NA	[^18^F]FDG-PET		Asymmetry indices (AIs) of cerebral metabolic rates for matched left-right regions of interest (ROIs) were calculated and the extent of diaschisis by correlative analyses was ascertained. For the Alzheimer group, cerebellar AIs correlated negatively, and thalamic AIs positively, with those of the cerebral hemisphere and frontal, temporal, parietal, and angular cortices, while basal ganglia AIs correlated positively with frontal cortical AIs. The only significant correlation of AIs for normal subjects was between the thalamus and the cerebral hemisphere. Crossed cerebellar as well as uncrossed basal ganglia and thalamic diaschisis in Alzheimer’s disease by positron emission tomography (PET) using 18F-fluorodeoxyglucose was detected in the study. These data indicate that PET is a sensitive technique for detecting diaschisis.	Retrospective
(Tanaka et al., 1993 [27])	26 (4 patients with dementia + Amylotrophic lateral sclerosis (ALS), 9 patients with ALS but without dementia, and 13 healthy.)	NA	NA	PET	Positron emission tomography with oxygen-15 gas and oxygen-15 labelled carbon dioxide	The mean regional cerebral blood flow (rCBF) and regional cerebral metabolic rate of oxygen metabolism (rCMRO2) in the anterior cerebral hemispheres decreased significantly in patients with progressive dementia with ALS, while patients with only ALS showed very mild reductions of rCBF and rCMRO2, which were not statistically significant in comparison to controls. Thus, the hypoperfusion and oxygen hypometabolism in the anterior cerebral hemispheres have an etiological relationship to cognitive impairment in patients with progressive dementia with ALS. A similar reduction in the mean rCBF was also found in the cerebellar hemispheres in progressive dementia with ALS, which was statistically significant, while a reduction in mean rCMRO2 was not significant. Remote effects analogous to crossed cerebellar diaschisis occurring bilaterally were assumed to explain the cerebellar hypoperfusion.	
(Al-Faham et al., 2014 [28])	1 old man	71 years	Michigan, USA	^18^FFDG-PET	After intravenous administration of of 18F-FDG, multiple tomographic slices of the brain were obtained with the 18F-FDG PET scan	An 18F-FDG PET scan demonstrated decreased radiotracer activity in the left and right parietal lobes, as well as left temporal and occipital lobes, which is compatible with Lewy body dementia. Moreover, there was decreased radiotracer uptake in the right cerebellum, suggestive of CCD.	Retrospective

## Data Availability

Not applicable.

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
