# Peer review of "A Systematic Review of PET Contrasted with MRI for Detecting Crossed Cerebellar Diaschisis in Patients with Neurodegenerative Diseases"

_diagnostics, 2023, doi:10.3390/diagnostics13101674_

Round 1
Reviewer 1 Report
Review
The authors aim to provide a systematic review of previous imaging studies detecting crossed cerebellar diaschisis in neurodegenerative disorders. Out of 273 published studies 8 retrospective studies were selected, of which 6 were positron-emission tomography (PET) studies, using F18-FDG-PET, 116 PiB PET or 18F-Flortaupicir, and 2 used magnetic resonance imaging (MRI) perfusion techniques, especially the arterial spin labelling perfusion MRI (ASIPWI). The latter is a perfusion technique used for quantitative arterial blood flow measurements without the need to inject contrast agents, using magnetically labelled blood water as an endogenous tracer. The result of the comparison is that crossed cerebellar diaschisis was better detected by functional imaging than by structural imaging. FDG-PET was most frequently used to detect the phenomenon of crossed cerebellar diaschisis.
Crossed cerebellar diaschisis refers to a decline in function, metabolism and perfusion affecting a cerebellar hemisphere, which occurs as a result of a contralateral focal supratentorial lesion. The most likely mechanism is thought to be the interruption of cortico-ponto-cerebellar white matter tracts, which then results in deafferentation and hypometabolism of the contralateral cerebellar hemisphere. Nuclear medicine must be applied to ascertain the diagnosis, and FDG-PET shows hypometabolism in the affected cerebellar hemisphere which is diagnostic of this phenomenon.
In the references, the authors cite articles which underline the bilateral symmetry of morphologic lesions in Alzheimer’s disease. They are unable to explain how a disease that generally affects brain such as Alzheimer’s, the most frequent and the most frequently investigated neurodegenerative disorder, can result in crossed cerebellar diaschisis, which occurs because of a focal unilateral supratentorial lesion on the contralateral side. Thus, the observation of a unilateral cerebellar depression in function, perfusion and metabolism without a contralateral brain lesion in the supratentorial area brings into question the mechanism and the existence of crossed cerebellar diaschisis in neurodegenerative disorders affecting the brain in a general rather than a distinctly asymmetrical manner.
The occurrence of unilateral cerebellar hypoperfusion and hypometabolism does not merit the term “crossed cerebellar diaschisis” lacking contralateral supratentorial region. In this sense, right from the title of the submitted article the term ”crossed cerebellar diaschisis” has already been misapplied.
The authors must either create a new concept of a unilateral depression in function, metabolism and perfusion of the cerebellum – so far well documented in several studies in neurodegenerative diseases, especially Alzheimer’s disease – lacking a contralateral supratentorial lesion, or they must coin a new term for the description. At the very least, they must address in greater depth the discrepancy between definition and description in their discussion. You cannot use a term in a scientific article which does not fulfil the preconditions of its definition: Crossed cerebellar diaschisis represents a depression of function, metabolism and perfusion affecting a cerebellar hemisphere and occurs as a result of a contralateral focal supratentorial lesion.
Author Response
Diagnosticss-ID-2318464
We would like to thank all the reviewers for their thoughtful comments and recommendations. We have carefully addressed all the suggestions and in doing so feel the manuscript is substantially strengthened. Attached are our detailed responses to their comments.
Reviewer #1:
The authors aim to provide a systematic review of previous imaging studies detecting crossed cerebellar diaschisis in neurodegenerative disorders. Out of 273 published studies 8 retrospective studies were selected, of which 6 were positron-emission tomography (PET) studies, using F18-FDG-PET, 116 PiB PET or 18F-Flortaupicir, and 2 used magnetic resonance imaging (MRI) perfusion techniques, especially the arterial spin labelling perfusion MRI (ASIPWI). The latter is a perfusion technique used for quantitative arterial blood flow measurements without the need to inject contrast agents, using magnetically labelled blood water as an endogenous tracer. The result of the comparison is that crossed cerebellar diaschisis was better detected by functional imaging than by structural imaging. FDG-PET was most frequently used to detect the phenomenon of crossed cerebellar diaschisis.
Crossed cerebellar diaschisis refers to a decline in function, metabolism and perfusion affecting a cerebellar hemisphere, which occurs as a result of a contralateral focal supratentorial lesion. The most likely mechanism is thought to be the interruption of cortico-ponto-cerebellar white matter tracts, which then results in deafferentation and hypometabolism of the contralateral cerebellar hemisphere. Nuclear medicine must be applied to ascertain the diagnosis, and FDG-PET shows hypometabolism in the affected cerebellar hemisphere which is diagnostic of this phenomenon.
In the references, the authors cite articles which underline the bilateral symmetry of morphologic lesions in Alzheimer’s disease. They are unable to explain how a disease that generally affects brain such as Alzheimer’s, the most frequent and the most frequently investigated neurodegenerative disorder, can result in crossed cerebellar diaschisis, which occurs because of a focal unilateral supratentorial lesion on the contralateral side. Thus, the observation of a unilateral cerebellar depression in function, perfusion and metabolism without a contralateral brain lesion in the supratentorial area brings into question the mechanism and the existence of crossed cerebellar diaschisis in neurodegenerative disorders affecting the brain in a general rather than a distinctly asymmetrical manner.
Response by authors: We thank this reviewer for their comments, and we agree with the reviewer that “Crossed cerebellar diaschisis refers to a decline in function, metabolism and perfusion affecting a cerebellar hemisphere, which occurs as a result of a contralateral focal supratentorial lesion and its occurrence in Alzheimer’s disease is not very well explained in the manuscript. We have explained and modified the manuscript accordingly. The occurrence of crossed cerebellar diaschisis in Alzheimer’s disease can be due to early involvement of cortico-ponto-cerebellar tracts in the course of the disease which was not very well explained earlier by the researchers as Alzheimer’s disease was thought to be predominantly a disease of cerebral hemisphere with cerebellar involvement very late in the disease process. Please the introduction, pages 4-7.
The occurrence of unilateral cerebellar hypoperfusion and hypometabolism does not merit the term “crossed cerebellar diaschisis” lacking contralateral supratentorial region. In this sense, right from the title of the submitted article the term ”crossed cerebellar diaschisis” has already been misapplied.
Response by authors: We greatly appreciate this reviewer for their comment, and we would like to highlight that the term crossed cerebellar diaschisis has been used for last a few decades in the literature for acute conditions such as stroke and focal lesion was mandatory for the same. However, lately, for a last few year’s researchers have started observing similar effect as contralateral cerebellar hypometabolism or hypoperfusion in response to hypometabolism or hypoperfusion in chronic neurodegenerative conditions such as Alzheimer’s disease.
The authors must either create a new concept of a unilateral depression in function, metabolism and perfusion of the cerebellum – so far well documented in several studies in neurodegenerative diseases, especially Alzheimer’s disease – lacking a contralateral supratentorial lesion, or they must coin a new term for the description. At the very least, they must address in greater depth the discrepancy between definition and description in their discussion. You cannot use a term in a scientific article which does not fulfil the preconditions of its definition: Crossed cerebellar diaschisis represents a depression of function, metabolism and perfusion affecting a cerebellar hemisphere and occurs as a result of a contralateral focal supratentorial lesion.
Response by authors: We thank this reviewer for their comments, and we would like to reiterate that contralateral cerebellar hypometabolism or hypoperfusion in response to hypometabolism or hypoperfusion in supratentorial cerebral hemisphere warrants same diagnostic entity as it is a well-established diagnosis and our understanding is further enhanced that it not only affects acute neurological condition but can also affect chronic neurodegenerative conditions.
Reviewer 2 Report
The study is aimed at generalizing information on the sensitivity and specificity of MRI and PET methods in the diagnosis of cerebellar diaschisis pathology. This is relevant because this pathology is clinically significant and the designation of the most effective method for the diagnosis of this pathology is necessary. The topic is not being considered for the first time, but the number of publications suitable for meta-analysis on this topic is not large enough. This article is an attempt to structure the data on the diagnosis of cerebellar diaschisis pathology by MRI and PET methods and choose the most appropriate one. The article is written well and clearly, the text is easy to read. The conclusions of the article are consistent with the presented arguments and evidence and answer the question of the optimal method of diagnosis of this pathology.Amyloidosis theory of the development of Alzheimer's disease has been criticized. It is necessary to additionally indicate other theories of the occurrence of the disease (immune system dysfunction or inflammation) with references to the authors in the text.
Author Response
Diagnosticss-ID-2318464
We would like to thank all the reviewers for their thoughtful comments and recommendations. We have carefully addressed all the suggestions and in doing so feel the manuscript is substantially strengthened. Attached are our detailed responses to their comments.
REVIEWER #2
The study is aimed at generalizing information on the sensitivity and specificity of MRI and PET methods in the diagnosis of cerebellar diaschisis pathology. This is relevant because this pathology is clinically significant and the designation of the most effective method for the diagnosis of this pathology is necessary. The topic is not being considered for the first time, but the number of publications suitable for meta-analysis on this topic is not large enough. This article is an attempt to structure the data on the diagnosis of cerebellar diaschisis pathology by MRI and PET methods and choose the most appropriate one. The article is written well and clearly, the text is easy to read. The conclusions of the article are consistent with the presented arguments and evidence and answer the question of the optimal method of diagnosis of this pathology.
Amyloidosis theory of the development of Alzheimer's disease has been criticized. It is necessary to additionally indicate other theories of the occurrence of the disease (immune system dysfunction or inflammation) with references to the authors in the text.
Response by authors: We thank this reviewer for their comments, and we have addressed the comments by the reviewer in the text. We have added a bit more information about other theories of occurrence of AD with appropriate references. Please see the introduction, pages 4-7.